# SVAD: From Single Image to 3D Avatar via Synthetic Data Generation with Video Diffusion and Data Augmentation

Yonwoo Choi
SECERN AI

yonwoo.choi@secern.ai
https://yc4ny.github.io/SVAD/

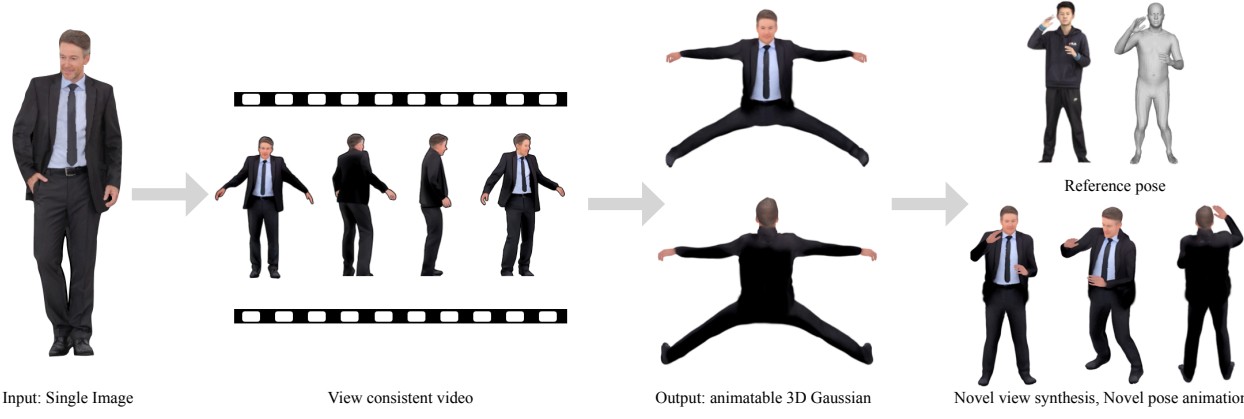

Figure 1. **SVAD.** Our method creates high-fidelity 3D avatars from a single image through synthetic data generation. We leverage video diffusion to generate pose-conditioned animations, enhance them with identity preservation and image restoration modules, then train a 3D Gaussian Splatting avatar. The resulting avatars maintain consistent identity across novel poses and viewpoints while enabling real-time rendering, outperforming state-of-the-art approaches.

## Abstract

*Creating high-quality animatable 3D human avatars from a single image remains a significant challenge in computer vision due to the inherent difficulty of reconstructing complete 3D information from a single viewpoint. Current approaches face a clear limitation: 3D Gaussian Splatting (3DGS) methods produce high-quality results but require multiple views or video sequences, while video diffusion models can generate animations from single images but struggle with consistency and identity preservation. We present SVAD, a novel approach that addresses these limitations by leveraging complementary strengths of existing techniques. Our method generates synthetic training data through video diffusion, enhances it with identity preservation and image restoration modules, and utilizes this refined data to train 3DGS avatars. Comprehensive evaluations demonstrate that SVAD outperforms state-of-the-art (SOTA) single-image methods in maintaining identity consistency and fine details across novel poses and viewpoints, while enabling real-time rendering capabili-*

*ties. Through our data augmentation pipeline, we overcome the dependency on dense monocular or multi-view training data typically required by traditional 3DGS approaches. Extensive quantitative, qualitative comparisons show our method achieves superior performance across multiple metrics against baseline models. By effectively combining the generative power of diffusion models with both the high-quality results and rendering efficiency of 3DGS, our work establishes a new approach for high-fidelity avatar generation from a single image input.*

## 1. Introduction

The ability to generate animatable 3D human avatars from minimal input data, such as a single-image, has significant potential across a range of applications. Traditional methods, particularly those based on 3DGS, have demonstrated considerable success in producing high-quality avatars [12, 27, 49, 50, 58, 66, 67, 73, 85, 92]. These methods rely on dense input data, typically monocular or multi-view

video [12, 27, 49, 58, 66, 73, 92], to achieve high fidelity across varied viewpoints and poses. This reliance on extensive video input complicates deployment in single-image scenarios, where ensuring viewpoint consistency and adaptability to novel poses becomes a key challenge.

Recent advancements in video diffusion models offer a potential solution by enabling animation generation from a single static image [26, 39, 77, 80, 93]. These models use certain conditions in diffusion processes to create video sequences, demonstrating the powerful generative capabilities of diffusion for single-image-driven animation. However, diffusion models often struggle to maintain temporal coherence, leading to inconsistent features and identity drift across frames [4, 16, 25, 70] . Additionally, their iterative denoising process for each frame introduces significant computational overhead, limiting their feasibility for real-time or interactive applications where rapid rendering across novel views is essential.

To overcome these challenges, we propose SVAD, a novel synthetic data generation and avatar creation pipeline that synergizes the generative flexibility of diffusion models with the efficient rendering capabilities of 3DGS avatars. Our approach leverages video diffusion model [74] to generate diverse pose-conditioned synthetic training data from a single-image. This synthetic data is refined through an identity-preservation module and an image restoration module to ensure that perceptual identity consistency and structural fidelity are preserved across diverse poses and temporal sequences. The resulting high-quality synthetic dataset is then used to train a 3DGS avatar model [49], which benefits from the rapid rendering capabilities inherent to 3DGS. By combining the generative strengths of diffusion for synthetic data creation with the efficiency of 3DGS for rendering, SVAD achieves consistent, high-quality 3D avatar animations from single-image input.

In summary, our main contributions are:

- We introduce a novel pipeline that generates high-quality synthetic training data from a single-image to create detailed, animatable 3D human avatars.
- We develop a comprehensive data augmentation approach that combines identity preservation and image restoration to ensure consistent identity and fine details across diverse poses.
- We demonstrate through extensive experiments that our synthetic data-driven approach significantly outperforms SOTA single-image avatar generation methods in identity preservation and novel pose adaptation while maintaining efficient real-time rendering.

## 2. Related Work

**Diffusion Model for Human Image Animation** The use of diffusion models has led to significant advancements in human image animation, enabling the generation of real-

istic and temporally consistent animations from static images [1, 5, 8, 18, 28, 33, 56, 60, 65, 68, 69, 82, 84, 87, 90]. Early methods, such as PIDM [3] and DreamPose [37], focused on improving texture fidelity by employing texture diffusion modules to align texture patterns between reference and target images. These methods, while enhancing detail preservation, still face challenges in maintaining temporal stability across frames.

Recent works, including DisCo [77] and Animate Anyone [26], have extended diffusion models to improve temporal consistency and fine-grained control in human animation tasks. DisCo leverages dual ControlNets [86] to separately control pose and background elements, providing more robust conditioning for complex motion sequences. Similarly, Animate Anyone integrates a ReferenceNet with temporal attention layers to ensure appearance consistency and smooth transitions across frames, thereby addressing flickering issues commonly observed in earlier models.

**Dynamic 3D Gaussian based Avatars** The concept of Gaussian splatting for 3D avatars has emerged recently as an innovative approach to explicit scene representation [38]. This technique models a scene as a collection of 3D Gaussian elements, each containing photometric and geometric properties. During rendering, these Gaussian splats are projected onto the image plane, creating the final rendered output. The efficiency of 3DGS has been demonstrated in both static [30, 35, 42] and dynamic [17, 36, 41, 43, 48] scenes, making it a versatile tool for various applications. Recent advancements [6, 7, 11, 15, 29, 32, 44, 57, 58, 76, 94] have explored the use of 3DGS to create photorealistic human avatars across different scenarios. These methods commonly rely on multi-view data [46, 53, 91] or monocular video [27, 32, 44, 49, 58] as input to achieve high-quality, consistent results. The advantage of 3DGS lies in its ability to produce temporally stable animated avatars with superior quantitative metrics.

## 3. Method

To generate high-quality human avatars from a single-image, facilitating free-viewpoint rendering and realistic animation, we integrate the generative capabilities of video diffusion models with the rendering efficiency of 3D Gaussian-based avatars. We start by leveraging a pretrained video diffusion model [74] for character animation to produce initial synthetic data, as described in Sec. 3.1. Directly using these frames to train a 3DGS avatar model [49], however, often yields poor results, with challenges in preserving facial identity, clothing details, and maintaining consistent multi-view coherence across side and back views. To address these issues and enhance avatar quality, we introduce a data augmentation pipeline in Sec. 3.2 comprising identity-preservation and image-restoration modules to refine the diffusion outputs. With the augmented synthetic

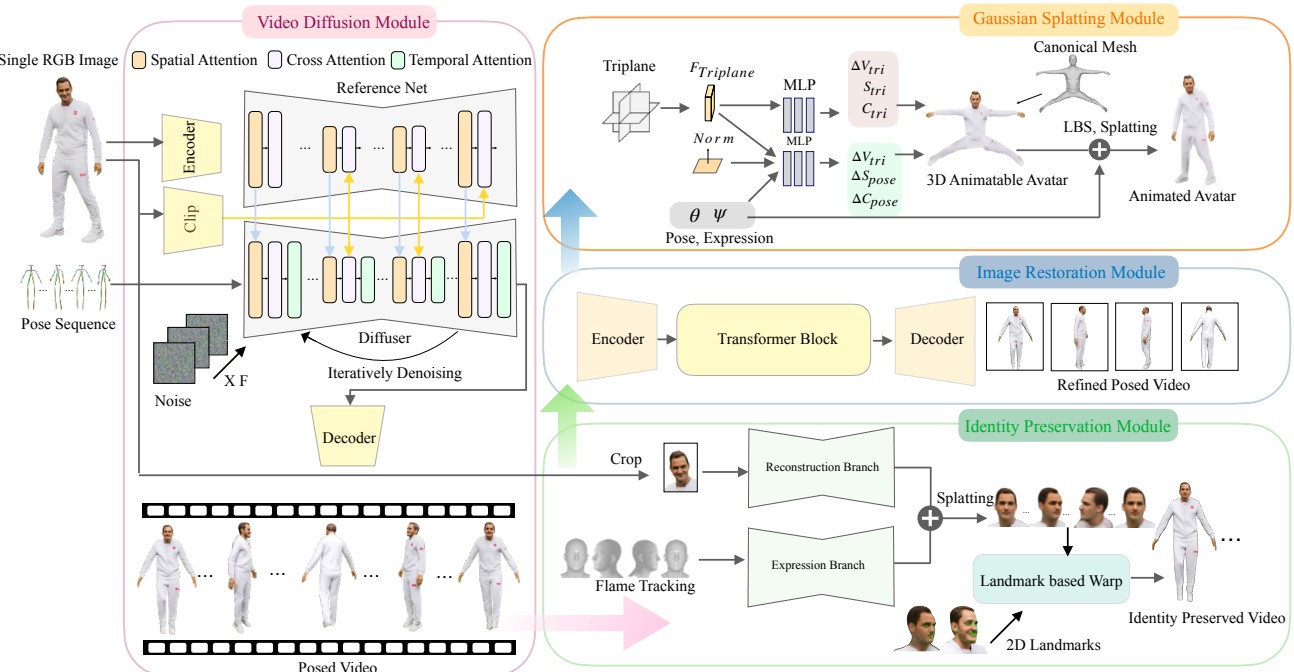

Figure 2. **Overall Pipeline of SVAD.** Starting from a single input image, the diffusion model generates pose-conditioned animations, which are refined using an identity preservation module and an image restoration module. The refined outputs are then used to train the 3DGS avatar, enabling high-fidelity, animatable 3D avatars with consistent details across poses and viewpoints.

data, we proceed to train a 3DGS avatar model, as outlined in Sec. 3.3. The following sections detail the technical methodologies employed in our approach.

## 3.1. Video Diffusion Module

To generate an animated character video $V$ from a single input image $I$, we leverage MusePose [74], a finetuned variant of Animate Anyone [26], which is a SOTA video diffusion model designed for realistic human animation while maintaining temporal consistency and appearance fidelity. MusePose employs a U-Net [62]-based diffusion architecture with integrated pose and temporal controls, allowing for pose-guided animation across frames. For our pipeline, we utilize a pose sequence video from a sequence from the People Snapshot [2] dataset, which depicts a subject performing a full-body rotation with arms extended horizontally. This sequence results in 189 frames that serve as pose inputs to the MusePose video diffusion model.

The model architecture incorporates several key components for effective character animation. The denoising UNet is implemented as a 3D UNet [14] with motion modules for temporal coherence. Specifically, we use Vanilla motion modules [20, 21] with temporal self-attention blocks at resolutions of [1, 2, 4, 8] and in the mid-block. Each transformer [75] block contains 8 attention heads, with temporal position encoding enabling positional awareness across

a sequence of up to 128 frames. To incorporate pose guidance, a lightweight Pose Guider encodes the motion control signal from the predefined 2D keypoints into a pose-aligned latent representation $P(p_t) \in \mathbb{R}^{H \times W \times C}$. For a pose feature $p_t \in \mathbb{R}^{J \times 2}$ at time $t$, where $J$ is the number of keypoints, we align the encoding to ensure continuity between frames by adding this encoded pose signal to the noise latent $z_t$:

$$z_t = z_t + P(p_t) \tag{1}$$

For the diffusion process, we adopt a v-prediction [64] formulation with zero-SNR sampling [47], using a scaled linear beta schedule with $\beta_{\text{start}} = 0.00085$ and $\beta_{\text{end}} = 0.012$. The DDIM [71] sampler is configured for efficient inference with 20 sampling steps and a classifier-free guidance [24] scale of 3.5.

A critical challenge in character animation is ensuring anatomical consistency between the reference image and the motion poses. Direct application of pose control can result in unnatural animations due to mismatches in body proportions [8]. Therefore, we employ a comprehensive pose alignment procedure that adapts the source pose to match the reference character's physical characteristics.

Given a reference pose $P_{ref}$ and a source pose $P_{src}$ detected using DWpose [81], we compute scale parameters $\mathbf{S} = \{s_1, s_2, \ldots, s_{10}\}$ for ten distinct body regions: neck, face, shoulders, upper arms, lower arms, hands, torso, up-

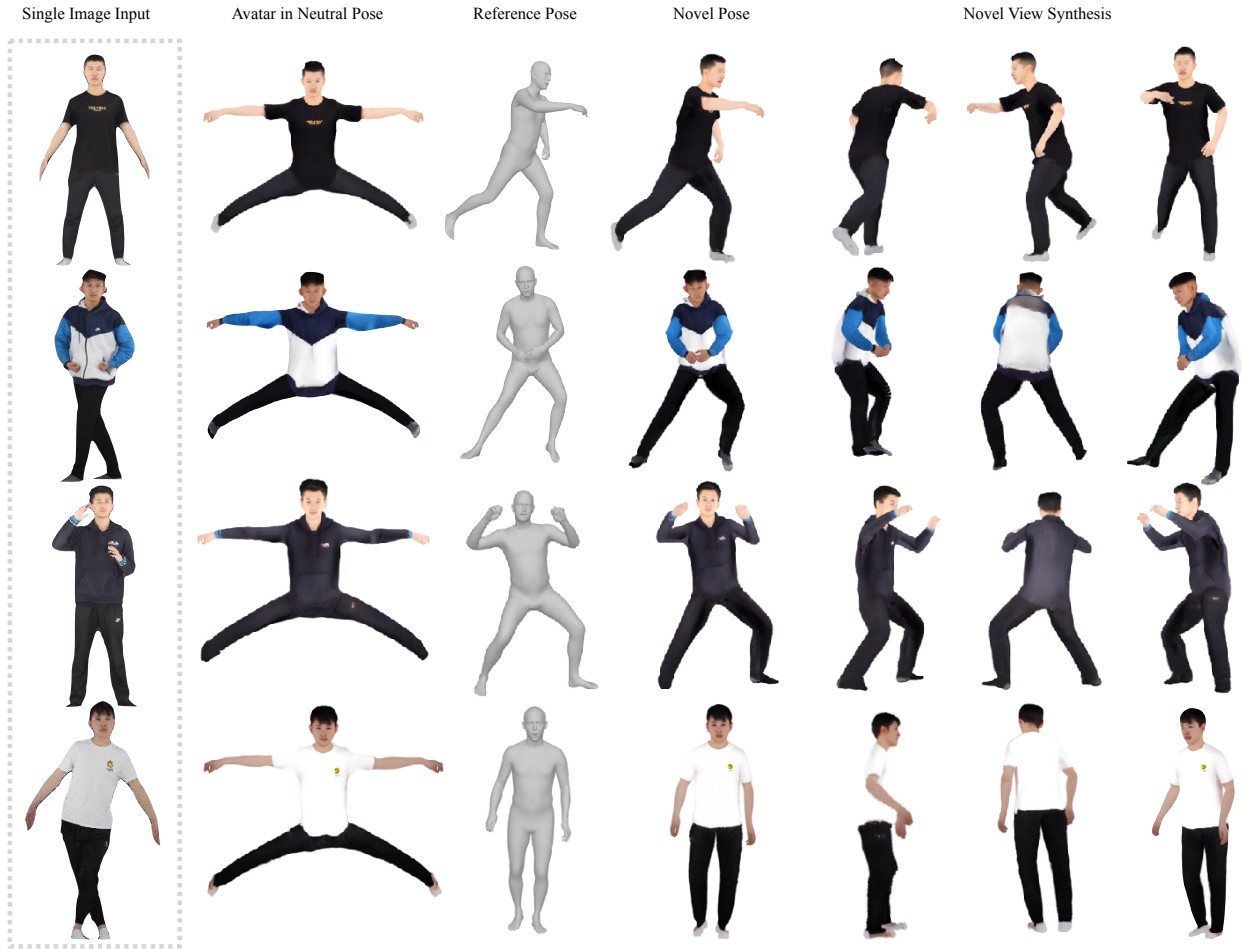

| Single Image Input | Avatar in Neutral Pose | Reference Pose | Novel Pose | Novel View Synthesis |
|---|---|---|---|---|

Figure 3. **3D Avatars trained by SVAD.** SVAD generates high quality 3D avatars with just a single-image. The trained avatars can be rendered from any view point, in any pose.

per legs, and lower legs. For each body part $i$, we compute its scale factor $s_i$ as the ratio between the corresponding keypoint distances. For body parts with bilateral symmetry (e.g., arms), we average the scales from both sides:

$$s_{arm\_upper} = \frac{1}{2} \left( \frac{\|p_{ref}^2 - p_{ref}^3\|}{\|p_{src}^2 - p_{src}^3\|} + \frac{\|p_{ref}^5 - p_{ref}^6\|}{\|p_{src}^5 - p_{src}^6\|} \right) \quad (2)$$

To apply these scales to the source pose, we use a rotation matrix transformation centered at anchor points specific to each body part:

$$p' = c_i + s_i \cdot (p - c_i) \quad (3)$$

where $c_i$ is the anchor center for part $i$. This hierarchical approach ensures body proportions match the reference while maintaining the overall pose structure.

### 3.2. Data Augmentation Module

Training the 3DGS model using only outputs from the video diffusion model often results in low-fidelity avatars, par-

ticularly in terms of facial details and high-frequency features like hands and clothing. To address these challenges, we introduce a data augmentation module that enhances the quality of the training data. This module includes an identity preservation sub-module ensuring coherence in facial details across frames and a image restoration submodule which refines texture quality and high-frequency details, resulting in more realistic textures. This comprehensive data augmentation significantly improves the synthetic training data, enabling the 3DGS avatar model integrated in the future to generate more realistic and detailed 3D avatars.

**Identity preservation sub-module.** To ensure consistent and realistic facial details across frames, we implement an identity preservation module that combines 3D head reconstruction and facial fusion techniques. From a single input image, we first create a 3D Gaussian-based head avatar using a method inspired by Chu *et al*. [13], which employs a novel *dual-lifting* approach that predicts both forward and backward lifting distances.

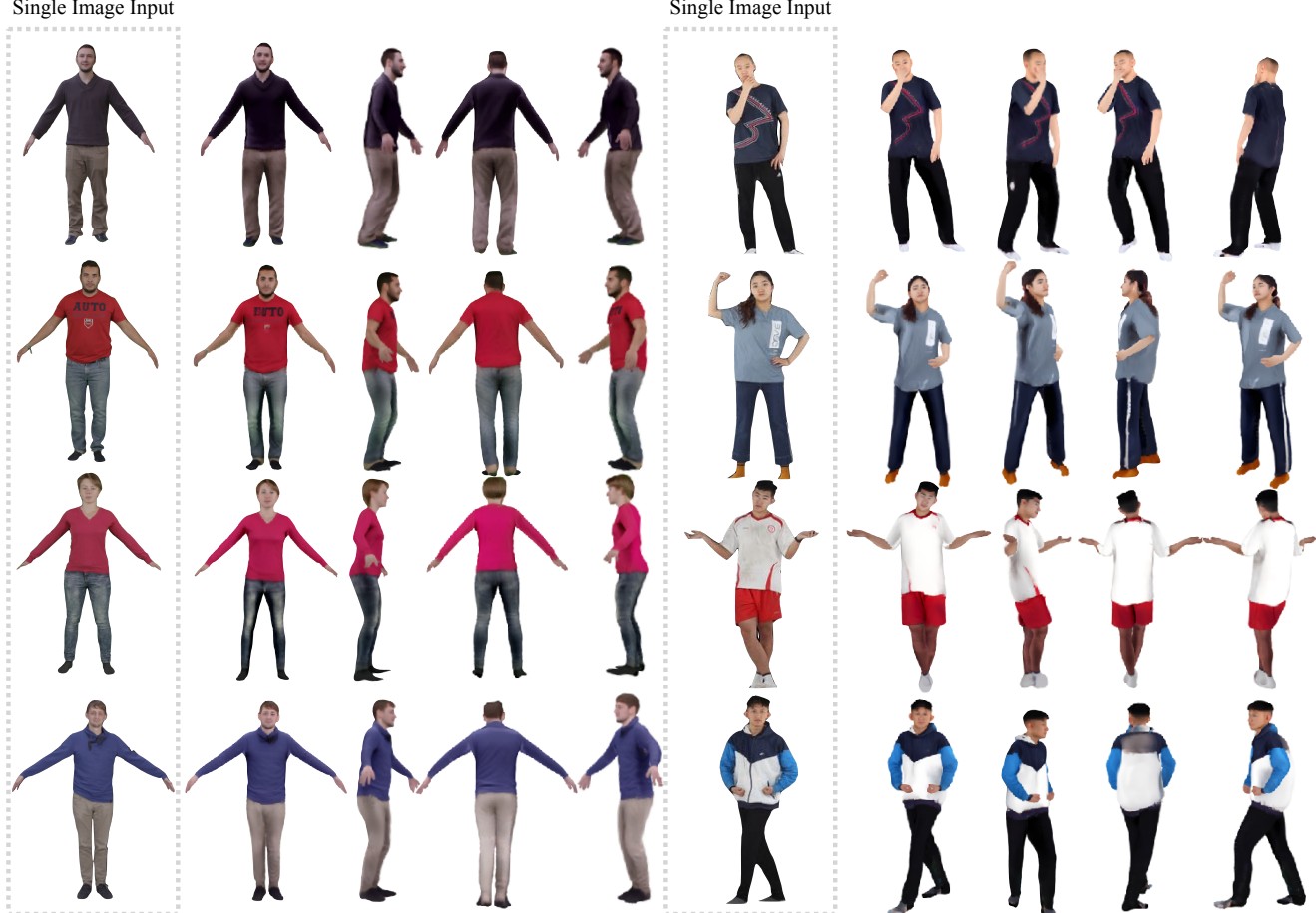

Figure 4. **Qualitative Evaluation** on the People Snapshot dataset and of THuman dataset scan renderings. From a single-image input, SVAD generates high-quality, animatable 3D avatars.

Given an input image $I_s$, global and local features $F_{local}$ are extracted using a frozen DINOv2 [52] backbone. These features are used to predict forward and backward lifting distances, positioning 3D Gaussians $G_{pos}$ as follows:

$$G_{pos} = [\mathbf{p}_s + E_{Conv0}(F_{local}) \cdot \mathbf{n}_s, \mathbf{p}_s - E_{Conv1}(F_{local}) \cdot \mathbf{n}_s],$$
(4)

where $\mathbf{p}_s$ is the initial point plane, $\mathbf{n}_s$ is the normal vector, and $E_{Conv}$ are convolutional layers predicting offsets. To capture expression variations, we bind 3DMM [45] features:

$$G_{expr} = MLP(F_{3DMM} + F_{global}).$$
(5)

To animate this 3D head avatar, we separately track FLAME [45] parameters $\Theta = \{\beta, \psi, \theta, \phi\}$ from our predefined pose sequence video (the same sequence used in the video diffusion module), where $\beta \in \mathbb{R}^{300}$ represents shape parameters, $\psi \in \mathbb{R}^{100}$ expression parameters, $\theta \in \mathbb{R}^6$ global pose parameters, and $\phi \in \mathbb{R}^6$ eye pose parameters. These tracked parameters serve as animation controls for

the reconstructed 3D head. Using these tracked FLAME parameters, we render the 3D head avatar to generate a sequence of head images that match our predefined pose sequence. These renderings provide high-quality, identity-consistent facial details across different viewpoints. Since the quality of the renderings deteriorates for back-of-head views, we selectively apply the face fusion process only to frames where the head is front-facing (front and side views).

For the face fusion process, we detect facial landmarks [40] on both the diffusion-generated frame $I_{orig}$ and the rendered head image $I_{head}$, compute an affine transformation for alignment, and use Poisson image editing [55] for seamless blending:

$$\min_I \int_\Omega \|\nabla I - \nabla I_{warp}\|^2 \, dx \, dy, \quad \text{subject to } I|_{\partial\Omega} = I_{orig}|_{\partial\Omega},$$
(6)

where $\Omega$ is defined by the facial mask. This ensures temporally consistent facial details while preserving the original identity throughout the animation sequence.

**Image restoration sub-module.** Finally, to preserve quality of fine detailed regions, we employ an image restoration module based on the work of Chen *et al.* [10], specifically their diffusion-based image restoration method BFRffusion. This approach leverages the generative prior encapsulated in the pretrained Stable Diffusion [61] model to enhance image details through a comprehensive architecture that effectively extracts features from low-quality images and restores realistic facial details.

For our implementation, we set the super-resolution scale factor to $s = 1.5$, which our empirical analysis showed provides an optimal balance between detail enhancement and artifact suppression. We observed that scale factors $s < 1.5$ produce insufficient detail recovery, while factors $s > 2.0$ introduce perceptual artifacts (particularly in specular regions such as eyes) and significantly increase computational demands during avatar training. The diffusion process uses 50 DDIM sampling steps with:

$$z_{t-1} = \sqrt{\alpha_{t-1}} \left( \frac{z_t - \sqrt{1 - \alpha_t}\epsilon_\theta(z_t)}{\sqrt{\alpha_t}} \right) + \sqrt{1 - \alpha_{t-1}}\epsilon_\theta(z_t) \tag{7}$$

where $\alpha_t = \prod_{i=1}^{t}(1 - \beta_i)$ and $\epsilon_\theta$ is the denoising network. We utilize a classifier-free guidance scale of $w = 3.5$, with the guidance equation:

$$\hat{\epsilon}_\theta(z_t) = (1 + w)\epsilon_\theta(z_t) - w\epsilon_\theta(z_t, \emptyset) \tag{8}$$

where $\epsilon_\theta(z_t, \emptyset)$ represents the unconditional prediction. This achieves an optimal balance between restoration quality and processing speed. For face regions, the method employs a face restoration helper with facial landmark detection to specifically enhance facial details, ensuring identity consistency across generated frames. Restored faces are blended with Poisson image editing.

This image restoration submodule significantly improves the fidelity and realism of our synthetic training data by restoring fine facial details, enhancing texture quality in clothing and accessories, and improving overall image coherence. The refined data enables the 3DGS avatar to learn more accurate representations with consistent high-frequency details that persist across poses and viewpoints.

### 3.3. 3D Human Gaussian Splatting Module

We apply the architecture of a 3DGS based avatar method introduced by Moon *et al.* [49], which integrates the SMPL-X [54] model with a 3D Gaussian-based representation to produce animatable human avatars. Each 3D Gaussian acts as a vertex connected by a pre-defined mesh topology following SMPL-X. This hybrid representation combines the expressive surface modeling of SMPL-X with the flexibility of a volumetric approach, allowing for smooth interpolation across the body surface essential for realistic animations.

Each Gaussian point is associated with positional data $\mathbf{V} \in \mathbb{R}^{N \times 3}$, RGB color values $\mathbf{C} \in \mathbb{R}^{N \times 3}$, and a scale parameter $\mathbf{S} \in \mathbb{R}^N$, where $N$ is the number of Gaussians. The Gaussian splatting rendering equation is:

$$I = f(V, \exp(S), C, K, E), \tag{9}$$

where $V$ represents positions, $S$ denotes scale, $C$ colors, and $K$ and $E$ camera parameters.

Pose-dependent deformations are applied through an MLP network, predicting offsets for each Gaussian based on SMPL-X pose parameters:

$$\mathbf{V}_{\text{pose}} = \mathbf{V} + \Delta\mathbf{V}_{\text{pose}} + \Delta\mathbf{V}_{\text{expr}}. \tag{10}$$

To maintain spatial coherence, a Laplacian regularizer [51, 72] minimizes the difference between the Laplacian of the canonical mesh and the deformed Gaussian points:

$$L_{\text{Lap}} = \|\Delta\mathbf{V}_{\text{canonical}} - \Delta\mathbf{V}_{\text{deformed}}\|^2. \tag{11}$$

This approach combined with our augmented synthetic data achieves highly realistic, animatable avatars capable of real-time rendering with smooth deformations across facial expressions, body movements, and hand gestures.

## 4. Experiments

### 4.1. Datasets and Metrics

**People-Snapshot Dataset** [2] We conduct our avatar evaluation on the People-Snapshot dataset, which features video recordings of subjects performing 360-degree rotations. Following both Anim-NeRF [79] and InstantAvatar [34], we address a known limitation in this dataset: the provided pose parameters often exhibit misalignment with the actual image content. Anim-NeRF addressed this by optimizing pose parameters for both training and test sequences. To ensure fair comparison with existing methods, we adopt these same optimized pose parameters and keep them frozen throughout our training process for fair comparison.

**THuman Dataset** [83] For evaluating single-image 3D human reconstruction, we employ the THuman dataset, adhering to the methodology established in Ultraman [9]. Our procedure involves randomly selecting 100 scans and generating renderings from four viewpoints (front, left, right, back). We then measure the similarity between our reconstructed outputs and the ground-truth scan renderings from these identical perspectives, facilitating objective comparison with other SOTA methods.

**Evaluation Metrics** Our evaluation framework uses four metrics to quantify reconstruction quality: PSNR [19], SSIM [78], LPIPS [88], and CLIP Similarity [59] (referred to as CLIP in our tables). This combination provides comprehensive assessment across different dimensions: PSNR for pixel accuracy, SSIM for structural coherence, LPIPS for perceptual alignment with human vision, and CLIP for semantic consistency at the feature level. The use of these metrics enables thorough evaluation of both fine-grained detail, and overall perceptual quality.

| Method | Female-4-casual | | | Male-3-casual | | | Female-3-casual | | | Male-4-casual | | |
|---|---|---|---|---|---|---|---|---|---|---|---|---|
| | PSNR↑ | SSIM↑ | LPIPS↓ | PSNR↑ | SSIM↑ | LPIPS↓ | PSNR↑ | SSIM↑ | LPIPS↓ | PSNR↑ | SSIM↑ | LPIPS↓ |
| HumanNeRF [79] | 27.07 | 0.9615 | 0.0151 | 26.90 | 0.9605 | 0.0181 | 24.46 | 0.9516 | 0.0269 | 25.50 | 0.9397 | 0.0357 |
| GaussianAvatar [27] | 30.84 | 0.9771 | 0.0140 | 30.98 | 0.9790 | 0.0145 | 29.55 | 0.9762 | 0.0220 | 28.78 | 0.9755 | 0.0230 |
| ExAvatar [49] | 30.98 | 0.9789 | 0.0333 | 29.75 | 0.9628 | 0.0402 | 29.74 | 0.9678 | 0.0458 | 28.89 | 0.9666 | 0.0500 |
| ExAvatar [49] (Single Image) | 20.42 | 0.9427 | 0.0656 | 23.24 | 0.9448 | 0.0562 | 20.12 | 0.9492 | 0.0543 | 23.74 | 0.9497 | 0.0610 |
| Ours (Single Image) | 21.51 | 0.9442 | 0.0528 | 22.54 | 0.9467 | 0.0484 | 21.96 | 0.9609 | 0.0541 | 23.71 | 0.9570 | 0.0592 |

Table 1. **Quantitative Evaluation** on the People Snapshot [2] Dataset. Our approach demonstrates superior performance on *single-image* input, outperforming the baseline on most of the metrics. The top two results for *single-image* input are highlighted in first and second, with the overall best result highlighted in first. Note that methods that use monocular input utilize approximately 200 input frames.

## 4.2. Quantitative Evaluation

We quantitatively evaluate the quality of single-image 3D avatars generated by our method against SOTA 3D avatar generation methods [27, 49, 79]. While current 3D avatar models generally require a monocular video as input, we assess our model's performance using a single-image as input on ExAvatar [49]. Additionally, we report results using the original full training set of approximately 200 input frames for monocular input based avatar models for reference. As shown in Table 1, our model achieves highest scores on most of the metrics among single-image input methods. We further compare our approach with single-view 3D human reconstruction methods [9, 23, 31, 63, 89], many of which employ the SMPL model, allowing for animatability through mesh fitting and reposing techniques, such as those in Editable Humans [22]. We randomly sample 100 scans from the THuman dataset and report results. We repose our trained avatar using ground-truth SMPL-X parameters and compare with the ground-truth scan renderings from the same views. As presented in Table 2, our method surpasses all baselines, demonstrating superior quality in 3D human reconstruction tasks.

## 4.3. Qualitative Evaluation

Figure 4 shows the overall quality of our generated 3D avatars from single-images in the People Snapshot and the THuman dataset. Figure 5, Figure 6 shows that our method performs superior compared to current SiTH [23]. For single-image avatar generation, we evaluate on the People Snapshot dataset and compare against ExAvatar [49]. For fairness, we train ExAvatar for the same number (12,000) of iterations. Figure 7 shows that for single-image avatar generation, our method performs superior especially for the back and side views.

## 4.4. Ablation Study

In this section, we conduct ablation studies to validate each component of our methods. The average metrics over 4 sequences in the People Snapshot dataset are reported in Table 3. It shows that our methods modules are required

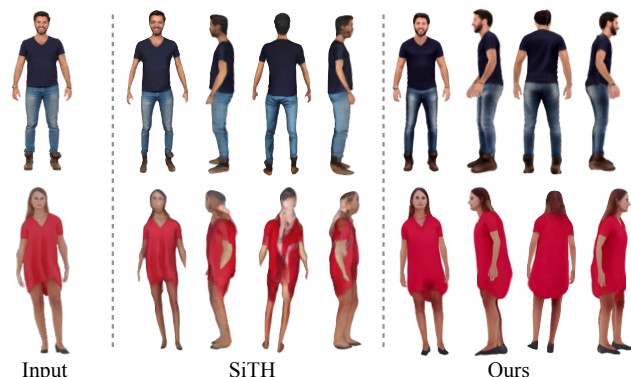

Figure 5. **Qualitative Evaluation** against SiTH [23]. Our approach better reconstructs complex contours and subtle features, resulting in a more lifelike and coherent side-view appearance.

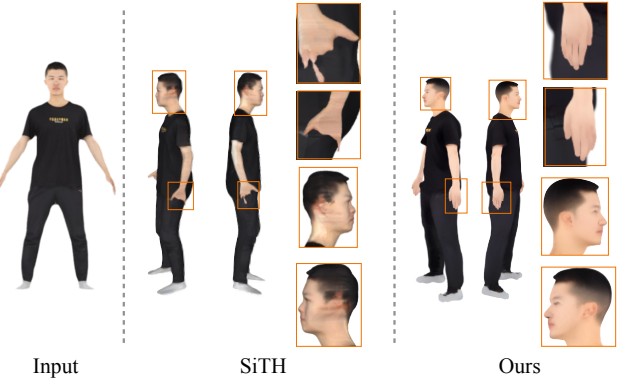

Figure 6. **Qualitative Evaluation** against SiTH [23]. Our method reconstructs fine detail (hands), while preserving original identity in facial regions.

to reach the optimal performance reflected by all the metrics. Using the THuman dataset, we apply the same evaluation technique as in our quantitative evaluation. Results show that our method performs the best in PSNR, SSIM and CLIP similarity and performs second best in LPIPS. Figure 8 shows visual results of the effect of the image restoration module. High-detailed regions such as clothing texture,

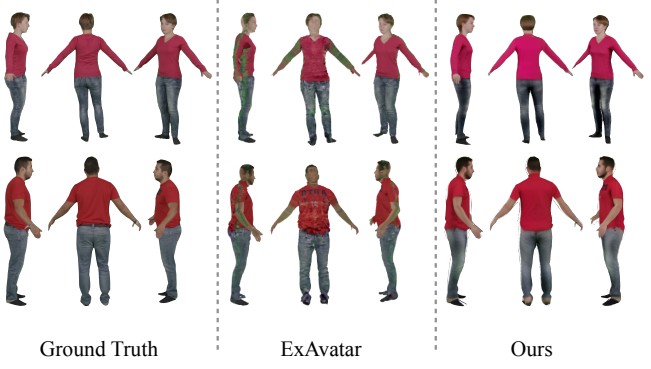

Ground Truth       ExAvatar       Ours

Figure 7. **Qualitative Evaluation** against ExAvatar [49] in single-image to 3D avatar task. Our method generates more plausible back and side views with the generated synthetic dataset.

| Method | PSNR↑ | SSIM↑ | LPIPS↓ | CLIP↑ |
|---|---|---|---|---|
| PIFu[63] | 15.62 | 0.8921 | 0.1903 | 0.8612 |
| TeCH[31] | 15.85 | 0.8892 | 0.1667 | 0.8890 |
| Ultraman[9] | 18.13 | 0.9019 | 0.1334 | 0.9089 |
| SIFU[89] | 18.59 | 0.8591 | 0.1402 | 0.8873 |
| SiTH[23] | 19.98 | 0.9018 | 0.1294 | 0.9084 |
| **Ours** | **20.92** | **0.9291** | **0.1124** | **0.9321** |

Table 2. **Quantitative Evaluation** on single-image to 3D human reconstruction tasks on 100 scan renderings of the THuman [83] Dataset. Top two results are colored as first second .

| Method | PSNR↑ | SSIM↑ | LPIPS↓ | CLIP↑ |
|---|---|---|---|---|
| w/o Identity Preserve | 22.19 | 0.9419 | 0.0623 | 0.9231 |
| w/o Image Restoration | 22.61 | 0.9298 | 0.0645 | 0.9239 |
| **Ours (Full)** | **22.79** | **0.9502** | **0.0594** | **0.9241** |

Table 3. **Ablation study** on the People Snapshot dataset. Our full model consistently outperforms variants with individual components removed across all metrics.

| Method | PSNR↑ | SSIM↑ | LPIPS↓ | CLIP↑ |
|---|---|---|---|---|
| w/o Identity Preserve | 20.12 | 0.9256 | 0.1294 | 0.9284 |
| w/o Image Restoration | 20.16 | 0.9212 | 0.0799 | 0.9201 |
| **Ours (Full)** | **20.92** | **0.9291** | **0.1124** | **0.9321** |

Table 4. **Ablation study** on the THuman dataset. The full model achieves superior performance in most metrics, demonstrating the importance of each component in our pipeline.

fingers, and facial details are better preserved when applying our module. Figure 9, shows the visual effect of the identity preservation module. We clearly show that original input's facial details are more preserved our module.

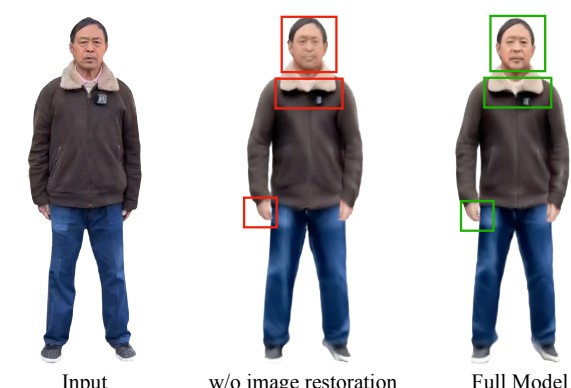

Input      w/o image restoration      Full Model

Figure 8. **Ablation study** on the image restoration module. We show that applying the module into our pipeline recover fine details on the final avatar output.

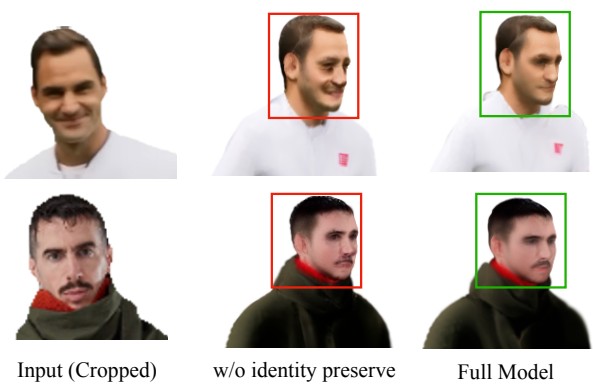

Input (Cropped)    w/o identity preserve    Full Model

Figure 9. **Ablation study** on the identity preservation module. We show that with the module, the final avatar maintains facial details on the original input image.

## 5. Conclusion and Discussion

In this work, we introduced SVAD, a novel synthetic data generation approach for creating high-fidelity, animatable 3D human avatars from a single image. By combining the generative power of diffusion models with the rendering efficiency of 3D Gaussian Splatting, SVAD produces avatars that maintain consistent identity across varied poses and viewpoints. Through comprehensive experiments, we demonstrate that our method achieves SOTA performance.

**Limitations and Future Work.** Our method faces several limitations. First, inaccurate background segmentation of training frames produces floating artifacts. Second, our approach struggles with complex clothing textures and loose outfits due to limitations of the video diffusion model in generating detailed synthetic data. Finally, the computational requirements present practical challenges—the video diffusion step demands substantial resources, and the complete pipeline requires 5-6 hours per avatar generation. Future work will focus on improving handling of diverse clothing types and optimizing computational performance.

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
