# SVAD: From Single Image to 3D Avatar via Synthetic Data Generation with Video Diffusion and Data Augmentation

## Supplementary Material

## Supplementary Material

This supplementary material provides additional details to complement the main paper. In Section 1, we elaborate on the implementation details of our pipeline, covering the predefined pose sequences utilized in our approach, the video diffusion module, the identity preservation module and image restoration module for enhancing facial fidelity and overall texture quality, and the fitting and training methodology for the 3DGS avatar module. Section 2 presents results on the data augmentation methods, highlighting their impact on improving the quality of the synthetic training data for the avatar model, with a focus on identity preservation and image restoration. In Section 3, we demonstrate the robustness of our method in handling challenging poses, including extreme body movements, occlusions, and non-frontal facial orientations. Section 4 discusses failure cases, identifying key limitations, while Section 5 outlines potential directions for future work aimed at addressing these challenges and further enhancing the robustness and realism of our approach.

## 1. Implementation Details

In this section, we provide comprehensive technical details of SVAD. We first describe the predefined pose sequences that serve as conditioning inputs for our video diffusion model. Next, we elaborate on the video diffusion module, the identity preservation module and image restoration module for enhancing facial fidelity and overall texture quality. Finally, we elaborate on the training process for our 3DGS avatar, including the SMPL-X [21] parameter fitting procedure and the optimization strategy for the 3D Gaussian representation.

### 1.1. Predefined Pose Sequences

To initialize frame generation for our pipeline, we rely on a predefined set of poses extracted from the People Snapshot [1] dataset. Specifically, we utilize the *male-4-casual* sequence, which depicts a subject performing a full-body rotation with arms extended horizontally. Using DWPose [32], we extract 2D keypoints $K \in \mathbb{R}^{J \times 2}$, where $J = 17$ is the number of keypoints, from this sequence to create a standardized pose template. This sequence serves as the conditioning input for the video diffusion model, resulting in 189 frames of pose-guided human animation, with a resolution of $1024 \times 1024$.

Our experiments revealed that inference with lower reso-

lutions such as $512 \times 512$ produced animations with significantly degraded facial details, which adversely affected subsequent processing steps. Particularly, the landmark-based face fusion technique requires accurate facial landmark detection, which proved unreliable on low-resolution outputs. The absence of distinct facial features in $512 \times 512$ outputs led to inconsistent landmark detection, compromising the accuracy of 3D head rendering and warping operations. The higher $1024 \times 1024$ resolution preserves critical facial details, enabling robust landmark detection and consistent face fusion results across the generated sequence.

### 1.2. Video Diffusion Module

For our video diffusion module, we leverage MusePose [27], a modified variant of Animate Anyone [12], specifically designed for pose-guided video generation from a single image. The architecture follows a UNet-based [24] denoising diffusion model with temporal modeling capabilities, enabling coherent video generation while maintaining consistency with the reference image.

During inference, the video diffusion pipeline performs iterative denoising of random noise guided by the reference image and pose sequence. We configure the DDIM sampler [26] with 20 sampling steps and a classifier-free guidance [10] scale of 3.5 which keeps balance between generation quality and inference speed. The network architecture employs a 3D variant of the standard UNet architecture, where temporal layers enable information exchange across video frames. The reference image features are extracted using a CLIP vision encoder [23] and processed through a reference UNet. These features are transferred to the denoising UNet via a custom attention mechanism:

$$\text{Attn}(Q, K, V) = \text{softmax}\left(\frac{QK^T}{\sqrt{d}}\right) V \qquad (1)$$

where $Q$ represents queries from the denoising UNet features, while $K$ and $V$ are derived from the reference image features. This mechanism ensures that generated frames maintain the appearance details of the reference image.

The pose conditioning is handled by the PoseGuider module, which processes pose skeleton images through a series of convolutional layers to create pose feature embeddings. These embeddings are added to the latent noise to spatially align the generation with target poses:

$$z_t = z_t + P(p_t) \qquad (2)$$

where $z_t$ is the noise latent at timestep $t$, $p_t \in \mathbb{R}^{J \times 2}$ is the pose feature at time $t$, and $P(\cdot)$ represents the pose

guider. The PoseGuider has an input convolutional layer, followed by blocks with increasing channel dimensions $(16, 32, 64, 128)$, and a zero-initialized output projection to the conditioning embedding channels.

For handling longer video sequences beyond the model's context window, we employ a sliding window [11] approach. The model processes frames in overlapping chunks of length $S = 48$ with an overlap of $O = 4$ frames. This enables the generation of arbitrarily long sequences while maintaining temporal consistency. The generative process for each video segment can be expressed as:

$$V_{i:i+S} = \mathcal{G}(I_{\text{ref}}, P_{i:i+S}, z) \tag{3}$$

where $V_{i:i+S}$ represents the generated video segment from frame $i$ to $i + S$, $\mathcal{G}$ is our diffusion model, $I_{\text{ref}}$ is the reference image, $P_{i:i+S}$ are the corresponding pose skeletons, and $z$ is the random noise. By processing these overlapping segments and blending them at the boundaries, the final full-length human-animated video has smooth transitions.

## 1.3. Identity Preservation Module

Following the initial frame generation by the video diffusion model, we refine the facial regions to enhance identity consistency and detail preservation. Our identity preservation pipeline consists of three main components: FLAME [18] parameter tracking 1.3.1, 3D head rendering 1.3.2, and face fusion 1.3.3. Each component plays a crucial role in generating high-quality, identity-consistent facial regions in our data augmentation pipeline.

### 1.3.1. FLAME Parameter Tracking

We begin by tracking FLAME parameters from our predefined pose sequence video to guide the animation of our 3D head avatar. Using a tracking engine with focal length set to 12.0, we extract parameters $\Theta = \{\beta, \psi, \theta, \phi\}$, where $\beta \in \mathbb{R}^{300}$ represents shape parameters, $\psi \in \mathbb{R}^{100}$ expression parameters, $\theta \in \mathbb{R}^{6}$ global pose parameters, and $\phi \in \mathbb{R}^{6}$ eye pose parameters.

To ensure smooth parameter transitions across frames, we apply Savitzky-Golay [14] filtering with a window length of 9 frames and polynomial order of 2. For rotation parameters, we employ quaternion-based smoothing [33] with a continuity enforcement algorithm to handle sign flips:

$$q'_{t+1} = \begin{cases} -q_{t+1}, & \text{if } q_t \cdot q_{t+1} < 0 \\ q_{t+1}, & \text{otherwise} \end{cases} \tag{4}$$

Different parameter types are smoothed with specific momentum coefficients: rotation matrices $\alpha = 0.6$, translation vectors $\alpha = 0.6$, and eye pose parameters $\alpha = 0.7$. This comprehensive smoothing strategy eliminates jitter and ensures temporal consistency in the final animation sequence.

### 1.3.2. 3D Head Rendering

Using GAGAvatar [5] as our 3D head modeling framework, we utilize the tracked FLAME parameters to render high-quality facial images that match our predefined pose sequence. We leverage this model to render the 3D head with precise control over pose and expression. The rendering process begins with the FLAME model, which generates 3D vertices based on the tracked shape, expression, pose, and eye parameters. We then employ a mesh renderer with a resolution of $512 \times 512$ pixels, using the FLAME topology for face modeling where focal length is set to 12.0. This approach enables us to generate precisely controlled facial renderings that maintain the identity of the source image while adopting the pose and expression parameters from the target sequence.

### 1.3.3. Face Fusion Process

We selectively apply face fusion only to frames when the head rendering is front-facing. We determine this by analyzing eye landmark detection - specifically, when at least one eye is clearly visible and properly detected in the facial landmark set. This approach ensures face fusion is only applied to frames with reliable facial orientation, as the quality of renderings deteriorates for back-of-head views where no eyes are visible. After filtering, we perform structural similarity assessment [29] and landmark-based warping [30] with careful parameter tuning to ensure seamless integration.

First, we detect 68 facial landmarks using dlib [15] on both the diffusion-generated frame $I_{\text{orig}}$ and the rendered head image $I_{\text{head}}$ from GAGAvatar. Before applying the transformation, we validate the structural compatibility by computing a Procrustes disparity measure [9] between the landmark sets:

$$d(L_{\text{orig}}, L_{\text{head}}) = \sqrt{\frac{1}{n} \sum_{i=1}^{n} \|L_{\text{orig},i} - L_{\text{head},i}\|^2} \tag{5}$$

where $L_{\text{orig}}$ and $L_{\text{head}}$ are the normalized landmark sets. We skip fusion when the disparity exceeds a threshold of 0.01, preserving the original frame in cases where the structural alignment would produce unnatural results. For valid frames, we compute an affine transformation matrix through corresponding landmarks using:

$$M = \arg\min_{M} \sum_{i=1}^{68} \|M \cdot L_{\text{head},i} - L_{\text{orig},i}\|^2 \tag{6}$$

where $M$ is a $2 \times 3$ affine transformation matrix. This matrix is estimated using a partial affine model that preserves scale while allowing for rotation and translation, maintaining proportional facial features during transformation. The

warped image is then computed by applying the transformation:

$$I_{\text{warp}} = T(I_{\text{head}}, M, (w, h)) \tag{7}$$

where $T$ represents the affine warping function that maps pixels from the source to destination image according to transformation $M$.

We then create a facial mask $\Omega$ by computing the convex hull [2] of the landmarks to define the facial region:

$$\Omega = \text{convexHull}(L_{\text{orig}}) \tag{8}$$

Finally, we apply seamless cloning, a gradient-domain blending implementation of Poisson image editing [22], centered at the face centroid $(c_x, c_y)$ with a blending factor $\alpha = 1.0$:

$$I_{\text{fused}} = \text{PoissonBlend}(I_{\text{warp}}, I_{\text{orig}}, \Omega, (c_x, c_y)) \tag{9}$$

This procedure solves the Poisson equation:

$$\min_I \int_\Omega \|\nabla I - \nabla I_{\text{warp}}\|^2 \, dx \, dy, \text{ subject to } I|_{\partial\Omega} = I_{\text{orig}}|_{\partial\Omega} \tag{10}$$

The gradient-domain blending preserves boundary conditions from the original image while replacing interior gradients with those from the warped image. This approach maintains lighting conditions and color consistency across the boundary by solving for pixel values that create a smooth transition while matching gradient fields. The complete face fusion pipeline significantly reduces visible artifacts at the transition between the rendered face and the original image, allowing consistent identity preservation even under challenging viewpoints.

## 1.4. Image Restoration Submodule

To enhance the quality of video diffusion outputs, particularly in facial regions, we integrate a hybrid restoration pipeline based on BFRffusion [4]. Our approach combines diffusion-based facial enhancement with background upsampling to improve overall visual fidelity while preserving identity-specific details.

The restoration workflow begins with face detection using RetinaFace [6], which accurately localizes facial regions in each frame. For aligned facial areas, we maintain a consistent face size of $512 \times 512$ with a $1:1$ crop ratio. When processing non-aligned faces, we employ a landmark-based alignment process using a five-point facial landmark detector with an eye distance threshold of $5$ pixels to filter out low-quality detections.

Each detected face undergoes diffusion-based restoration using a latent diffusion model. The process follows a conditional diffusion sampling approach:

$$z_{t-1} = \frac{\sqrt{\alpha_{t-1}}z_t - \sqrt{1 - \alpha_t}\epsilon_\theta(z_t)}{\sqrt{\alpha_t}} + \sqrt{1 - \alpha_{t-1}}\epsilon_\theta(z_t) \tag{11}$$

where $\alpha_t = \prod_{i=1}^t (1 - \beta_i)$ and $\epsilon_\theta$ is the denoising network. We implement classifier-free guidance with a scale of $w = 3.5$:

$$\hat{\epsilon}_\theta(z_t) = (1 + w)\epsilon_\theta(z_t) - w\epsilon_\theta(z_t, \emptyset) \tag{12}$$

where $\epsilon_\theta(z_t, \emptyset)$ represents the unconditional prediction.

The diffusion sampling process uses 50 DDIM steps with a latent shape of $\mathbb{R}^{4 \times 64 \times 64}$ for $512 \times 512$ input images. The input facial image is first encoded to a latent representation through a VAE encoder, and the diffusion model progressively refines this representation before decoding it back to pixel space.

For background regions, we employ Real-ESRGAN [28] with an RRDBNet [8] architecture and a $2\times$ upsampling scale. The background upsampler processes images in tiles of $400 \times 400$ pixels with 10-pixel padding to handle high-resolution inputs efficiently while maintaining consistent quality across tile boundaries.

After separate processing of facial and background regions, we integrate the enhanced components using inverse affine transformations computed from the original facial alignment process. This creates a seamless composite where facial details are preserved and enhanced while maintaining natural transitions to background areas:

$$I_{\text{final}} = M_{\text{face}} \odot T^{-1}(I_{\text{face}}) + (1 - M_{\text{face}}) \odot I_{\text{bg}} \tag{13}$$

where $T^{-1}$ represents the inverse transformation that maps the restored face back to its original position, and $M_{\text{face}}$ is the binary mask indicating facial regions.

This comprehensive image restoration approach significantly enhances the perceptual quality of generated frames, particularly improving fine facial details that may be lost or degraded during the initial video diffusion process. The integration of specialized facial and background processing ensures optimal quality across the entire frame while maintaining computational efficiency.

## 1.5. Gaussian Avatar Submodule

To transform our synthetic data into a high-quality, animatable 3D avatar, we employ a two-stage process: first, we fit an SMPL-X model to our synthetic data sequences, then we train a 3D Gaussian Splatting representation using the fitted parameters as guidance.

### 1.5.1. SMPL-X Fitting Process

Prior to training the 3DGS avatar, we employ a comprehensive fitting process to obtain accurate SMPL-X parameters from our synthetic data. This multi-stage process ensures that the avatar's geometry accurately reflects the subject's physical characteristics and articulation.

**Keypoint Extraction.** The fitting pipeline begins with pose and shape estimation. We utilize DWPose [32] to extract 2D

whole-body keypoints from each frame of our synthetic sequence. These keypoints provide critical information about body articulation across the sequence. The keypoints are represented as $K \in \mathbb{R}^{J \times 3}$, where $J = 133$ includes 17 body, 68 face, and 42 hand keypoints, with each keypoint having $(x, y, \text{confidence})$ values. We then employ MM-POSE [25] with the RTMPose-L [13] model for refinement, using a confidence threshold of 0.5 to filter reliable detections.

**Initial Parameter Estimation.** For facial geometry, we leverage DECA [7] to estimate initial FLAME parameters. The optimization uses perspective projection with focal length of 5000 pixels and $1024 \times 1024$ resolution textures. The FLAME parameters include shape coefficients $\beta \in \mathbb{R}^{10}$, expression parameters $\phi \in \mathbb{R}^{10}$, and pose parameters for jaw and eyes.

For body pose and shape, we incorporate Hand4Whole [19] with the configuration: focal length of 2000, principal point at image center, and input shape of $256 \times 256$. This process yields initial estimates for SMPL-X parameters: global orientation $\theta_{\text{root}} \in \mathbb{R}^3$, body pose $\theta_{\text{body}} \in \mathbb{R}^{21 \times 3}$, jaw pose $\theta_{\text{jaw}} \in \mathbb{R}^3$, hand poses $\theta_{\text{hands}} \in \mathbb{R}^{30 \times 3}$, and shape parameters $\beta_{\text{shape}} \in \mathbb{R}^{10}$.

**Parameter Optimization.** These initial parameters are refined through an optimization process with multiple objectives. The primary loss function combines reprojection error, parameter regularization, and temporal smoothness:

$$L_{\text{fit}} = \lambda_{\text{kpt}} L_{\text{kpt}} + \lambda_{\text{reg}} L_{\text{reg}} + \lambda_{\text{temp}} L_{\text{temp}} \tag{14}$$

The keypoint reprojection loss $L_{\text{kpt}}$ measures the distance between projected model joints and detected 2D keypoints, weighted by detection confidence:

$$L_{\text{kpt}} = \sum_{i=1}^{J} c_i \| \Pi(J_i(\theta, \beta)) - K_i \|_2^2 \tag{15}$$

where $\Pi$ is the perspective projection function, $J_i(\theta, \beta)$ is the 3D position of joint $i$, $K_i$ is the corresponding 2D keypoint, and $c_i$ is its confidence score.

The regularization term $L_{\text{reg}}$ penalizes deviation from prior pose and shape distributions:

$$L_{\text{reg}} = \| \beta \|_2^2 + \sum_j \| \theta_j - \theta_{\text{mean}} \|_2^2 \tag{16}$$

The temporal consistency term $L_{\text{temp}}$ enforces smooth transitions between frames:

$$L_{\text{temp}} = \sum_{t=1}^{T-1} \| \theta_t - \theta_{t+1} \|_2^2 + \| \beta_t - \beta_{t+1} \|_2^2 \tag{17}$$

The optimization uses the Adam optimizer [16] with learning rate $1 \times 10^{-3}$ and loss weights $\lambda_{\text{kpt}} = 1.0$, $\lambda_{\text{reg}} =$

0.001, and $\lambda_{\text{temp}} = 0.1$. The optimization proceeds in two stages: first optimizing global position and orientation with 100 iterations, then refining all parameters with 200 iterations.

**Parameter Smoothing.** To ensure temporal consistency and reduce jitter, we apply the same smoothing approach as used in our FLAME parameter tracking process in Section 1.3.1. Specifically, we employ Savitzky-Golay [14] filtering with a window length of 9 frames and polynomial order of 2. For rotation parameters, we utilize the identical quaternion-based smoothing procedure with continuity enforcement to handle sign flips.

**Segmentation and Depth Estimation.** We generate foreground masks using Segment Anything [17] with the ViT-H backbone. The model uses keypoint-based prompting with valid keypoints as point coordinates, and a bounding box computed from these keypoints with an extension ratio of 1.2. We also extract depth information using Depth Anything V2 [31] with the ViT-L backbone. The depth maps are normalized and aligned with the SMPL-X mesh using the following procedure:

$$\text{scale} = \frac{\sigma(\text{depth}_{\text{pred,fg}})}{\sigma(\text{depth}_{\text{smplx,fg}})}$$

$$\text{depth}'_{\text{pred}} = \frac{\text{depth}_{\text{pred}}}{\text{scale}}$$

$$\text{depth}'_{\text{pred}} = \text{depth}'_{\text{pred}} - \mu(\text{depth}'_{\text{pred,fg}}) + \mu(\text{depth}_{\text{smplx,fg}}) \tag{18}$$

where $\sigma$ and $\mu$ represent standard deviation and mean of depth values, and fg indicates foreground regions.

The extracted SMPL-X parameters $\Phi = \theta, \beta$, together with corresponding image observations $I_t t = 1^T$, foreground masks $M_t t = 1^T$, and aligned depth maps $D_t{}_{t=1}^T$, constitute a multi-modal conditioning set that guides the optimization of our 3D Gaussian representation.

### 1.5.2. 3DGS Avatar Training Process

With the fitted SMPL-X parameters and processed synthetic data, we proceed to train the 3DGS-based avatar [20]. The training begins by initializing the triplane representation [3] $T \in \mathbb{R}^{32 \times 128 \times 128}$, encoding 3D features for both body and facial regions. Gaussian parameters, including positions $\mathbf{V} \in \mathbb{R}^{N \times 3}$, colors $\mathbf{C} \in \mathbb{R}^{N \times 3}$, and opacity $\mathbf{O} \in \mathbb{R}^N$, are optimized through backpropagation with the following multi-objective loss function:

$$L = \lambda_{\text{RGB}} L_{\text{RGB}} + \lambda_{\text{SSIM}} L_{\text{SSIM}} + \lambda_{\text{LPIPS}} L_{\text{LPIPS}}, \tag{19}$$

where $\lambda_{\text{RGB}} = 0.8$, $\lambda_{\text{SSIM}} = 0.2$, and $\lambda_{\text{LPIPS}} = 0.2$ are the weights for the RGB reconstruction, structural similarity, and perceptual loss, respectively. The model is trained for 5 epochs with a batch size of 1, as required by the Gaussian splatting renderer.

The optimization process proceeds in two stages. During the warmup stage, Gaussian positions $\mathbf{V}$ are updated using an adaptive learning rate:

$$\alpha_{\text{position}}(t) = \alpha_{\text{init}} \times \left(1 - \frac{t}{T_{\max}}\right) + \alpha_{\text{final}} \times \frac{t}{T_{\max}}, \quad (20)$$

where $\alpha_{\text{init}} = 1.6 \times 10^{-4}$, $\alpha_{\text{final}} = 1.6 \times 10^{-6}$, and $T_{\max} = 30,000$ iterations. Additional parameters, including opacity $\mathbf{O}$, scale $\mathbf{S}$, and feature parameters, are optimized with learning rates $\alpha_{\text{opacity}} = 0.05$, $\alpha_{\text{scale}} = 0.005$, and $\alpha_{\text{feature}} = 0.0025$, respectively.

Densification of the Gaussian distribution occurs between iteration $500$ and $15,000$, at intervals of $100$ iterations. Gaussians with opacity values below a threshold ($\mathbf{O} < 0.005$) are pruned, and dense regions are refined using gradient-based adjustments. The pruning mechanism ensures efficient representation while preserving fidelity:

$$\mathbf{V}_{\text{new}} = \mathbf{V}_{\text{old}} - \eta \frac{\partial L}{\partial \mathbf{V}}, \quad (21)$$

where $\eta$ is the learning rate and $\frac{\partial L}{\partial \mathbf{V}}$ represents the gradient of the loss with respect to Gaussian positions.

A hierarchical learning approach progressively increases the spherical harmonic degree $d_{\text{sh}}$ from $0$ to $3$ over the course of training. The training loop dynamically adjusts Gaussian parameters, leveraging an Adam optimizer with a learning rate of $1 \times 10^{-3}$ for the overall framework and parameter-specific rates for finer control. For our experiments, we employ the male SMPL-X [21] model due to its superior performance in complex sequences. The entire pipeline runs on a single GPU, ensuring scalability and efficiency.

## 2. Details on Generated Synthetic Data

We show our augmented data from the sequences of the People Snapshot [1] dataset. As shown in Fig. 1, applying our data augmentation module consisted of the identity preservation and the image restoration sub-module enhance the overall quality of the data, especially the facial regions.

## 3. Challenging Poses

As shown in Fig. 2, our method shows robustness to challenging poses, including extreme body movements, occlusions, and non-frontal facial orientations. This robustness is achieved through the integration of pose-guided video diffusion models and the 3D Gaussian splatting framework, which together enable high-fidelity avatar generation that remains consistent across a wide range of poses and motions. The ability to handle such diverse and dynamic poses is critical for applications requiring realistic and adaptable avatar rendering. The capability to handle such challenging poses and motion scenarios establishes the robustness

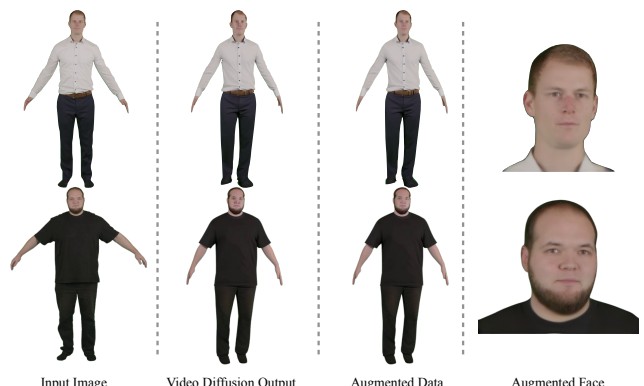

Input Image     Video Diffusion Output     Augmented Data     Augmented Face

Figure 1. **Data Augmentation Results.** This figure highlights the effectiveness of our data augmentation pipeline, showcasing enhanced facial regions and overall image quality improvements achieved through the identity preservation and image restoration sub-modules

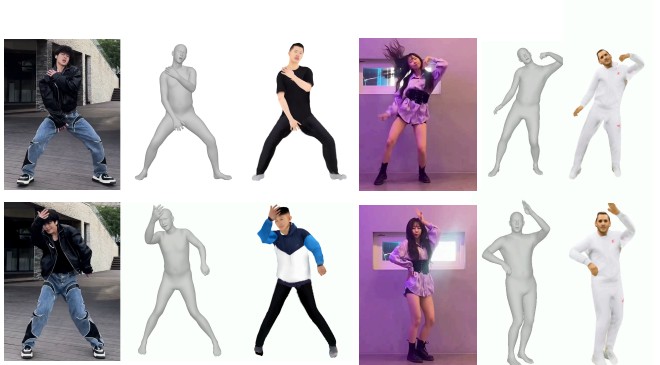

Figure 2. **Challenging Poses.** The figure illustrates the robustness of our method in handling extreme poses, including non-frontal views and dynamic motion scenarios, while maintaining high fidelity and consistency in the generated avatars.

and versatility of our method, making it well-suited for applications in gaming, virtual reality, and animation. Future enhancements, such as incorporating additional motion datasets and refining pose-guidance mechanisms, could further extend this capability to even more complex and dynamic scenarios.

## 4. Failure Cases

In this section, we analyze several failure cases observed in SVAD, revealing limitations in specific scenarios that highlight areas for potential improvement. As in Fig. 3 one of the primary challenges lies in the generation of back and side views. The inherent bias towards frontal views within diffusion models often results in noisy or inaccurate reconstructions of non-frontal regions. These inconsistencies are particularly evident in textured areas, such as clothing and hair, where fine details are difficult to maintain without

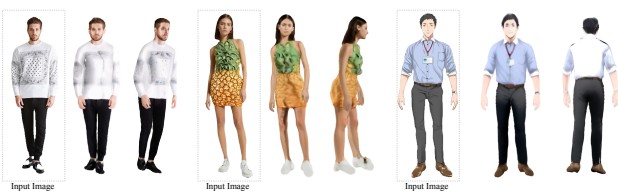

Figure 3. **Failure cases of SVAD.** Examples include noisy back and side views, inconsistent clothing textures, and artifacts in non-frontal regions.

multi-view constraints.

Another issue arises in maintaining consistent textures and lighting across different viewpoints. Artifacts such as abrupt transitions in lighting or shading irregularities appear, particularly in side or back views. These imperfections likely stem from limitations in the data augmentation process, as synthesized views may not fully capture the diversity of real-world lighting conditions and texture variations. These inconsistencies affect the overall visual fidelity and reduce the photorealism of the rendered avatars.

Additionally, while the 3D Gaussian splatting representation is effective for free-viewpoint rendering, its reliance on isotropic Gaussians can lead to oversmoothing in high-frequency regions such as hands and facial features. This limitation occasionally causes a loss of sharpness and detail in regions where fine textures are crucial for realism.

## 5. Future Work

Addressing these limitations in Sec. 4 requires several enhancements. Improvements to the data augmentation pipeline, such as introducing more realistic texture and lighting variations, could help mitigate shading and texture artifacts. Regularization techniques could enforce more consistent geometry and appearance across views, while hybrid volumetric representations or pose-dependent deformation fields could improve accuracy in challenging poses. These advancements would help SVAD achieve greater robustness and fidelity across diverse scenarios.