# OpenReview forum: "SVAD: From Single Image to 3D Avatar via Synthetic Data Generation with Video Diffusion and Data Augmentation"
_thecvf.com/CVPR/2025/Workshop/SyntaGen — SyntaGen 2025 Poster_

### Official Review · Reviewer_hFyh · 2025-03-25

**Rating:** 5
**Confidence:** 3

**Review:**

This paper introduce SVAD, a novel pipeline that leverages video diffusion models and data augmentation methods to generate high
quality synthetic training data from a single human image. Specifically, the pipeline starts with a video diffusion model fine-tuned with human poses, generating raw posed video. After that, the multi-view images from the posed animations are refined by identity preservation module and and image restoration module.

This paper is well-written and easy-to-follow with a newly proposed dual-lifting approach for identity preservation. However, I have two major concerns: (1) The quantitative results in Table 1 look unsatisfied; (2) Only the qualitative comparisons with SITH [1] and ExAvatar [2] are given, which is not sufficient.

Weighing the strengths and weaknesses, I would recommend a rating of 5.

*References*
[1] SITH, Ho et al,  Sith: Single-view textured human reconstruction with image-conditioned diffusion, CVPR 2024
[2] ExAvatar, Moon et al, Expressive whole-body 3d gaussian avatar, ECCV 2024

---

### Official Review · Reviewer_DDm4 · 2025-03-26
**This paper presents an interesting idea that leverages video diffusion models to generate synthetic data, along with a data augmentation module to improve the quality of the generated video before using it to train 3D human avatars given only a single image. However, the paper could be further improved, as mentioned in the Weaknesses and Additional Comments sections. The core idea of the paper has some merit for the workshop.**

**Rating:** 7
**Confidence:** 3

**Review:**

### **Paper Summary:**
SVAD presents a pipeline that leverages video diffusion models to generate synthetic data for training 3D human avatars from a single image. This pipeline consists of three key components: the video diffusion module, the data augmentation module which enhances the generated video’s image quality and reduces identity shift and the 3DGS training process for human avatars. This paper achieves state-of-the-art (SOTA) performance in the single-image-based setting and provides ablation studies to demonstrate the importance of each component.

### **Paper Strengths:**
1. The proposed key idea of leveraging the video diffusion model to generate video is very interesting and simple.
2. The proposed pipeline is designed in a model-agnostic way, meaning it is not restricted to any specific model and can potentially be improved as newer video diffusion models emerge.
3. As shown in the ablation study, the identity preservation module and the image restoration module help improve the raw output of the video diffusion model by addressing identity shift issues and enhancing poorly generated areas (e.g., facial parts, clothing).

### **Paper Weaknesses:**
On the other hand, the paper has the following weeknesses:
1. In Table 1, comparing with ExAvatar in the single-image scenario might introduce unfairness, as ExAvatar requires a monocular video as input rather than a single image. Similarly, in the qualitative results (L346–L349), comparing ExAvatar by training it on a single image may not be fair to the state-of-the-art methods, as this does not align with its original input requirements.
2. Some of the implementation details that used in evaluation are missing:
    - The minimum #frames to be generated from video diffusion model.
    - The minimum requirement for variation of poses that required to generate the results (e.g., 360 degree rotation or -30 to 30 rotation from frontal facing).


### **Justification For Recommendation And Suggestions:**
This work introduces an interesting idea, and its design is easy to apply to any new video diffusion model. However, the paper could be further improved, as mentioned in the Weaknesses and Additional Comments sections.

### **Additional Comments For Authors:**
1. There is potentially a typo in L336, where the table reference should point to Table 2 for the comparison with single-image 3D human reconstruction, also the current text is missing a link to the results in Table 2.
2. From the results in Table 1, this method might not surpass other baselines that require a monocular video as input, especially since it aims to leverage a video diffusion model from a single image. This comparison between single-image and monocular-video-based methods may be unfair, as they target different problem setups. To strengthen the evaluation, one could show a comparison using the same input frames as the state-of-the-art methods, with and without the additional generated frames, to determine whether the performance can be improved. This could potentially support the narrative of using synthetic data to enhance model performance.
3. For the formula (2), it's unclear whether this loss function used in which module, is it used during finetuning the video diffusion model, training it from scracth or during denoising process?

---

### Official Review · Reviewer_gxHK · 2025-03-27
**SVAD review**

**Rating:** 6
**Confidence:** 5

**Review:**

Summary: the paper tackles the problem of generating animatable avatar from a single image by leveraging video diffusion model and data augmentation. From the pose sequence and the input image, the authors first generate a posed video of that person. The quality of the posed video is later enhanced by two modules: one for identity preservation and one for overall enhancement. Finally, the method uses Exavatar to create an animatable 3d avatar from the enhanced pose video.

Strength:
- The method is able to generate animatable 3d avatar that can render novel views and poses which is impressive.
- The paper is well-written and easy to follow. The proposed method of using data augmentation seems like a well fit for this workshop.

Weakness:
- The proposed methods are combination of well-known existing works.
- The method relies on video diffusion model which is compute extensive.
- Despite having two enhancement modules, the animatable 3D avatar is still low quality and blurry.

---

### Decision · Program_Chairs · 2025-03-30

**Decision:**

Accept (Poster)

**Comment:**

The paper received Marginal Accept, Accept, and Marginal Reject. The last reviewer raised concerns about the unsatisfactory results in Table 1 and noted that the qualitative comparisons, provided only for [1, 2], were insufficient. However, the reviewer did not elaborate why they were unsatisfactory or insufficient. The other two reviewers found the animatable 3D avatar that can render novel views and poses impressive and considered the ideas interesting. The ablation study also demonstrates improvements from individual pipeline modules. Despite the noted weaknesses, the ACs agreed with the overall positive impression and felt that the contribution is a good fit for the workshop.